# eHealth Literacy and Knowledge of Social Health Financing Among Undergraduate Healthcare Students in Kenya

**DOI:** 10.3390/ijerph22101560

**Published:** 2025-10-13

**Authors:** Elham Aldousari, Maha Alhajeri, Dennis Kithinji

**Affiliations:** 1Department of Health Informatics and Information Management, College of Allied Health Sciences, Kuwait University, Sulaibikhat 90805, Kuwait; maha.alhajeri@ku.edu.kw; 2Department of Medical Laboratory Sciences, College of Health Sciences, Meru University of Science and Technology, Meru 60200, Kenya; 3Department of Research and Writing, Medright Consulting Ltd., Maua 60600, Kenya

**Keywords:** health literacy, information literacy, digital media, insurance, health, national health insurance, healthcare financing, students, health occupations, Kenya

## Abstract

Low rates of actively contributing to the Social Health Insurance Fund (SHIF) under the Social Health Authority (SHA) could be due to health insurance knowledge inadequacies, possibly because of poor electronic health (eHealth) literacy. This study assesses whether eHealth literacy is associated with SHA/SHIF knowledge among undergraduate healthcare students in Kenya. An analytical cross-sectional study was conducted using the eHealth Literacy Scale (eHEALS) and an adapted Kaiser Family Foundation quiz. A total of 207 students in mainly six health-related academic programs in 21 institutions of higher learning in Kenya responded to the online survey. Only 54% and 21.7% of the participants had high (median ≥ 4 out of 5) levels of eHealth literacy and knowledge of SHA/SHIF, respectively. About 9.2% of the students had never heard of SHA/SHIF. Only high eHealth literacy compared to low eHealth literacy (OR = 6.2, *p* < 0.0001) and pursuing nursing, public health, and other programs compared to pursuing Bachelor of Medicine and Bachelor of Surgery (MBChB) (OR = 4.9–5.1, *p* = 0.01–0.03) significantly predicted SHA/SHIF knowledge. Thus, eHealth literacy levels and SHA/SHIF knowledge require improvement among undergraduate healthcare students in Kenya to prepare them as SHA ambassadors in their communities.

## 1. Introduction

The Kenyan government launched the Social Health Authority (SHA), comprising Social Health Insurance Fund (SHIF); Primary Healthcare Fund (PHP); and Emergency, Chronic, and Critical Illness Fund (ECCIF) in October 2024, with premiums funding SHIF and taxes funding PHF and ECCIF [1]. SHA is a transition from the allegedly inequitable National Hospital Insurance Fund (NHIF) into a more inclusive and universal health coverage (UHC)-oriented healthcare financing model [2]. Unfortunately, voluntary contributions into SHIF are low as Kenyans in the informal sector remain inactive members of SHA.

More than 19 million Kenyans were enrolled for SHA as of February 2025, indicating a considerable reach [3]. However, only 3.5 million people, primarily the formally employed, were actively contributing to SHIF since the premium is automatically deducted from their salaries [4]. Kenyans in the informal sector may be avoiding SHA due to consumption of inaccurate SHA information from online sources as a result of low eHealth literacy levels.

The eHealth literacy level of an individual is a combination of traditional, health, information, scientific, media, and computer literacies as per the Lily Model of eHealth Literacy; literacy that determine the use of online tools and navigation of apps as per the Transaction Model of eHealth literacy and the Digital Health Literacy Model; and literacy from interacting with user content in social media platforms as per the Health 2.0 Literacy Frameworks [5]. Liu et al. highlighted the critical role that eHealth literacy plays in the sustainability of structured healthcare models [6]. Njuguna and Wanjala asserted that investing in health awareness initiatives contributes to the broader dissemination of accurate health insurance information, which increases public trust and participation in health insurance systems [7]. Therefore, identifying intervention areas for improving eHealth literacy levels and health insurance knowledge is vital in the efforts to ensure every Kenyan consistently contributes to SHIF.

Increasing awareness about health insurance and specifically SHA can boost voluntary contributions to SHIF. The government of Kenya leverages the growing digital infrastructure and social media use to promote SHA, even though similar initiatives in other African countries did not adequately address disinformation and a lack of clarity about health policy details [2,8]. Influencing the public with health information depends on people’s eHealth literacy, which enhances searching, retrieving, appraising, synthesizing, and correctly applying online health information for better health decision-making [9]. For example, a Swedish cross-sectional analysis by Ahlstrand et al. found that high eHealth literacy among university students is associated with better health decision-making and greater engagement with digital health information [10]. Similarly, Diviani et al. caution that limited awareness of credible online health information sources, which is attributed to low eHealth literacy, exposes digital health consumers to a high risk of invalidated information that may be false or misinterpreted [11]. University students in healthcare academic programs presumably have high eHealth literacy levels compared to the general population due to their existing knowledge of health concepts [11,12]. Hence, they are expected to be better informed on health insurance compared to the general public.

Advanced knowledge of health insurance and eHealth literacy among undergraduate healthcare students in Kenya is crucial in SHA awareness strategies since families and communities of the students trust them for health information. Barasa et al. and Mugo agreed on the necessity of public health education initiatives to enhance health literacy and bridge the knowledge gaps that marred NHIF use [13,14]. As the reliance on digital communication for SHA awareness and information dissemination increases, undergraduate healthcare students can be used as the focal points to improve accessibility, retrieval, interpretation, and use of online health insurance information by communities in Kenya. Their health sciences background, community trust, and eHealth literacy can be leveraged to ensure that the general public, especially people in the informal sector, have access to the information they need to make informed decisions on contributing to SHIF.

If the students enrolled in healthcare academic programs are prepared through contextualized health insurance and eHealth literacy training, they could be low-cost focal points to educate their community members about health insurance to address the problem of limited knowledge of the social health insurance benefits package [15,16]. This is because every village in Kenya most likely has an undergraduate student in health sciences who can be equipped with better eHealth literacy to make them champions for social health insurance uptake through awareness. They require the least resources to train into eHealth literacy-equipped SHA crusaders since they already have the basic health insurance knowledge, digital health exposure, and community goodwill [17]. Unfortunately, a literature review did not identify studies estimating the levels of eHealth literacy and knowledge of health insurance of undergraduate healthcare students in Kenya and how they relate. This study aimed to assess the levels of and association between eHealth literacy and SHA knowledge of undergraduate healthcare students pursuing health programs in Kenya toward identifying intervention areas to contextually prepare them to be SHA crusaders in their communities, especially in the informal sector, where health insurance uptake is low.

## 2. Materials and Methods

Study design: Analytical cross-sectional study.

Target and accessible population: Undergraduate students pursuing healthcare academic programs in Kenyan universities were the target population. The accessible population comprised undergraduate healthcare students on major social media platforms in Kenya.

Sample size estimation: The sample size was 120 estimated using G*Power 3.1.9.7 software with the following input parameters: test family = z rests; statistical test = logistic regression; tails = 2; odds ratio = 2; probability of health insurance knowledge = 25%; significance level (α) = 0.05; power = 0.8; multicollinearity adjustment = 0.2; median rating = 3 (neutral); and variance = 1.

Ethical considerations: This study involved human participants; hence, it was conducted while adhering to the principles of the Declaration of Helsinki on the ethical conduct of research involving human participants, including informed consent and confidentiality. No personal identifiable information, contact details, or IP addresses were collected during the survey to protect the respondents’ privacy. MUST Institutional Research Ethics Review Committee (MIRERC) reviewed the protocol for this study and approved it (MIRERC 018/2025). The National Commission for Science, Technology, and Innovation (NACOSTI) granted research license number NACOSTI/P/25/418223 for this study after reviewing the protocol.

Sampling: Self-selection, a non-probabilistic sampling technique, was applied. People who came across the survey link online and felt motivated, interested, and available to complete the survey clicked the link to participate.

Recruitment: One researcher and research assistants across universities in Kenya shared social media posts inviting undergraduate healthcare students to participate in the online survey and indicating the survey link on TikTok, X, Instagram, Facebook, LinkedIn, and class emails. Additionally, the post was promoted on WhatsApp status to reach people aged 18 to 27 years, which is the age group for most undergraduate healthcare students in Kenya.

Variables: The dependent variable was knowledge of social health insurance—SHA and SHIF in Kenya. The main independent variable was eHealth literacy. Other independent variables included gender, institution of study, program of study, and year of study.

Data collection tools: The survey tool comprised a demographics section, an eight-item eHealth literacy scale, and a 10-item adapted Kaiser Family Foundation (KFF) Health Insurance Knowledge Quiz. The eHealth literacy scale is widely validated and has a Cronbach’s alpha of 0.8 based on psychometric analysis among internet-based college students [18], which shows good internal consistency and reliability for use among university students accessible through social media platforms. It is self-reported, whereby participants indicate their perceived self-confidence and comfort in retrieving and applying health information from electronic sources in a five-item Likert scale (from strongly disagree to strongly agree). The KFF Health Insurance Knowledge Quiz is user-friendly, and it elicits reliable responses that represent a unidimensional construct of health insurance knowledge from university students [19], which indicates good validity. It tests knowledge of the main health insurance concepts and terms using multiple-choice questions, hence revealing overall knowledge level and specific knowledge gaps of the specified health insurance program.

Data collection: The survey was distributed through Google Forms to potential respondents as a survey link accessible through https://forms.gle/YEauT4cwpjMcnSAHA (made accessible on 31 April 2025). The survey responses were viewed in a Google sheet and downloaded into Excel for analysis.

Data analysis: Data were sorted, cleaned, and coded for analysis using SPSS Version 25. Data were summarized using median, interquartile range, and frequencies. Reliability analyses were conducted to obtain Cronbach’s alpha for the eHealth literacy scale and the Health Insurance Knowledge Quiz. Man Whitney U test (two groups) and Kruskal–Wallis test (more than two groups) were conducted to determine whether eHealth literacy levels and SHA/SHIF knowledge differed across groups since the ratings data were ordinal. Post hoc Dunn’s test with Bonferroni correction was conducted when at least one group’s median rating was identified as statistically significantly different in the Kruskal–Wallis test. Spearman’s rank correlation measured the direction and strength of association between eHealth literacy and SHA/SHIF knowledge, controlling for age, gender, year of study, and socio-economic status. Penalized logistic regression identified predictors of SHA/SHIF knowledge.

## 3. Results

Demographic characteristics.

A total of 207 undergraduate healthcare students responded to the survey between 31 April 2025 and 31 May 2025. Their mean age was 21.7 years (standard deviation = 2.1) while the age range was 18–31 years. Male and female students were almost equally represented (Table 1). The participants were studying in 21 institutions of higher learning, comprising 12 public institutions and 9 private institutions spread across Kenya.

Six major and twelve other undergraduate healthcare programs offered in Kenya were represented, with medicine, nursing, and medical laboratory science (MLS) topping the list. The other academic programs include pharmacy, dentistry, physical therapy, and microbiology.

Reliability of scales.

Both the eHealth literacy scale and the SHA and SHIF knowledge scale had excellent reliability, with Cronbach’s alpha of 0.9 (95% CI: 0.88–0.92) and 0.96 (95% CI: 0.95–0.97), respectively. The inter-item correlations of the items ranged from 0.51 to 0.57 for the eHealth literacy scale and 0.70–0.72 for the SHA and SHIF knowledge scale, both indicating strong inter-item consistency.

eHealth literacy levels.

The median total eHealth literacy score was 31 (Q1 = 28, Q3 = 33). The range of the total scores was 32 (minimum = 8; maximum = 40). Only 53.6% (n = 111) of them had high (median ≥ 4) eHealth literacy levels, with 5.3% (n = 11) having low (median ≤ 2) and 41.1% having modest (median = 3) eHealth literacy levels. Comparisons of total eHealth literacy scores across gender, university, and academic program did not reveal any statistically significant differences.

However, comparisons of the total eHealth literacy scores across years of study revealed that at least one of the years had a different median from the rest (Kruskal–Wallis H = 12.4, df = 3, *p* = 0.006). Post hoc Dunn’s test with Bonferroni correction showed that only senior students’ eHealth literacy scores (median = 4) were statistically significantly different from third year students’ (median = 3.75, *p* = 0.02) and first year students’ (median = 3.75, *p* = 0.02), which may not translate to practical differences.

Out of the eight individual items of the eHealth literacy scale, only knowledge of how to use the internet and find helpful health resources online was rated highly (Figure 1).

Knowledge of SHA.

Only 9.2% (*n* = 19) of the 207 participants had never heard of SHA/SHIF. The median total score of knowledge of SHA was 30 (Q1 = 22, Q3 = 36), with the minimum and maximum total scores being 10 and 50, respectively. Only 21.7% (*n* = 45) of the participants had a high (median ≥ 4) level of knowledge of SHA; 30.4% had low (median ≤ 2), while 47.9% had modest knowledge of SHA (Figure 2).

A binary logistic regression was conducted to determine whether age, gender, institution, academic program, and year of study predicted having heard of SHA/SHIF. The logistic regression model was statistically significant for the academic program (χ^2^ (1) = 9.9, *p* = 0.002) and age (χ^2^ (1) = 5.0, *p* = 0.03). However, only the academic program of study statistically significantly (*p* = 0.004) predicted having heard of SHA/SHIF (OR = 1.45, 95% CI [1.1, 1.9]), with students in MBCHB, nursing, and MLS having higher odds of having heard of SHA compared to their counterparts in community health. Similarly, only comparisons across academic programs showed existence of students with a statistically significant total score of SHA knowledge (Kruskal–Wallis H = 14.7, df = 5, *p* = 0.012). However, post hoc Dunn’s test with Bonferroni correction showed that none of the academic programs’ learners were statistically significantly different in terms of SHA knowleddge from the learners of the other individual academic programs.

Association between eHealth literacy and knowledge of SHA.

eHealth literacy total scores were statistically significantly positively correlated with the total scores of the knowledge of SHA (Spearman’s rho = 0.36, *p* = 0.0001). The SHA knowledge level and eHealth literacy level variables were recoded into two measures: low (a combination of low and modest) and high (retained as it was) to conduct binary logistic regression. Collinearity diagnostics were conducted using linear regression. The distribution of the SHA knowledge level (high or low) in relation to age, gender, institution, academic program, year, eHealth literacy level, and having heard of SHA was predicted using penalized logistic regression (Firth method) instead of binary logistic regression. This is because although variance inflation factor (VIF) values ranged from 1.0 to 1.6 and tolerance values ranged from 0.6 to 1.0, showing that levels of collinearity among the predictors were acceptable, cross-tabulation revealed that frequencies of the outcome were more than 10 in some categories of the predictors.

All the included predictors except age were categorical. The overall logistic regression model was statistically significant, χ^2^ (17) = 43.15, *p* = 0.0005. Only eHealth literacy and partaking in some academic programs were significant predictors of SHA knowledge level (Figure 3).

The respondents with high eHealth literacy were six times more likely to have a high knowledge level of SHA compared to those with low eHealth literacy levels, controlling for age, gender, university, academic program, year of study, and having heard of SHA (Table 2). Similarly, compared to the respondents studying MBChB, nursing, public health, and other academic programs, students were approximately five times more likely to have a high level of SHA knowledge, controlling for all the other predictors in the model.

## 4. Discussion

eHealth literacy levels.

About half of Kenya’s undergraduate healthcare students have suboptimal eHealth literacy levels, indicating that they may not be effectively differentiating credible information from gutter content about SHA [20]. A similar study among young adults in four universities in Kenya reported a median eHealth literacy rating of 3.21 out of 5 [21], which is slightly lower than the 3.88 observed in this study. In Ethiopia, two studies, one among all undergraduate healthcare students and the other specifically among nursing University students, reported mean ratings of 3.58 and 3.15, respectively [13,22]. Therefore, undergraduate healthcare students have slightly higher eHealth literacy levels compared to other university students, but the levels are suboptimal considering the high rates of smartphone and internet penetration in Kenya.

eHealth literacy could be improving as undergraduate healthcare students transition from year 1 to years 4, 5, and 6, but it remains suboptimal even in senior years of study, indicating the need to supplement the health curricula in Kenya with eHealth literacy concepts. Year 4, 5, and 6 students have advanced education levels and age, which are known factors that influence eHealth literacy levels [22,23,24]. eHealth can be taught in tertiary education for healthcare workers of the future to enhance their readiness to work in an environment characterized by abundant eHealth information and technologies [25]. Therefore, if eHealth is integrated into the curricula of undergraduate healthcare students in Kenya, their eHealth literacy levels can progressively improve more significantly as they transition from first year to fourth, fifth, and sixth years.

The undergraduate healthcare students had suboptimal eHealth literacy levels in reliability evaluation, quality assessment, application, and discussion of online health information, and application of digital technology and knowledge of the available online health resources. Competencies to evaluate reliability of eHealth information are critical since the overall quality of health information is concerning [26]. Undergraduate healthcare students should also have the capacity to assess the quality of online health information, as for-profit organizations and individuals marketing products and services embrace health information-sharing as a marketing strategy [27]. Since people are projected to increasingly seek health information online [28], healthcare professionals in training should prepare to discuss the credibility of various information sources with patients and community members to optimize their online health-seeking behaviors. Therefore, the undergraduate healthcare students need improvements in all eHealth aspects.

Knowledge of SHA.

Nearly 10% of the undergraduate healthcare students self-reported having never heard of SHA or SHIF, yet the government has been conducting both mainstream media and digital media sensitizations, showing that information penetration was subpar. NHIF awareness in the general population was 81.5% in four counties in Western Kenya in 2019 [29], which aligns with the current findings since health insurance knowledge levels are expected to be higher among undergraduate healthcare students compared to the general population in Kenya, which mainly comprises people in the informal sector [30]. Knowledge of health insurance determines the rate of its uptake [31]. Consequently, having 10% of trusted health advocates and prospective healthcare workers not knowing about health insurance means that the unawareness is likely to cascade to the community members who depend on them for health information [32]. Since receiving health insurance instructions is associated with better knowledge and self-efficacy about health insurance [33], the national health insurance instructions can be integrated into the academic programs of undergraduate healthcare students in Kenya to ensure that all of them have the knowledge.

Having only a fifth of undergraduate healthcare students in universities in Kenya with high levels of SHA knowledge indicates a worrying situation, considering that the proportion could be even smaller in the general population. A similar study in Nigeria reported high levels of knowledge of a university health insurance system among students in a Nigerian university, but the detailed understanding of the services covered by the insurance was low [34], indicating superficial knowledge. The high likelihood of SHA/SHIF knowledge among MBCHB, nursing, and MLS students compared to other undergraduate healthcare students could be due to their more intense interaction with the healthcare system in Kenya during practicums, resulting in better knowledge. However, learning from the clinical environments is unreliable since it depends on the approachability and supportiveness of clinical supervisors [35]. Accordingly, health insurance education should be integrated into the coursework component of healthcare workers’ training for system sustainability and equity [36], considering Kenya’s diverse clinical practicum settings.

Undergraduate healthcare students with health insurance knowledge are more likely to embrace advocacy about health insurance, considering the knowledge–persuasion–decision–implementation–confirmation continuum of spreading ideas [37]. In addition, health insurance literacy is associated with greater utilization of preventive and primary care services, which can result in better health outcomes overall [38]. Learning from a Public Health Youth Ambassador Program that equipped students as community health workers in Virginia, USA [39], Kenyan undergraduate healthcare students’ health insurance literacy can be optimized alongside equipping them with science communication skills to strategically position them as SHA ambassadors to support plain-language awareness initiatives among community health promoters and the general public in their local communities.

Association between eHealth literacy and knowledge of SHA.

The positive correlation between eHealth literacy and knowledge of SHA reveals an opportunity to improve knowledge of SHA using eHealth literacy interventions. High eHealth literacy levels compared to low eHealth literacy levels were the main predictor of high levels of knowledge of SHA. High eHealth literacy is associated with better online health-seeking behaviors [21], which is likely to increase exposure to information about SHA. A cohort study conducted among Kenyatta University students in the school of business and the school of education reported that students whose phones had a health-related mobile app were more likely to have high levels of health awareness [40]. Undergraduate healthcare students with high eHealth literacy levels can use the most recent evidence-based online materials not only to boost their health insurance knowledge but also to empower community members to make informed decisions on health insurance [41]. Therefore, incorporating eHealth literacy into the curricula of healthcare academic programs in Kenya can enhance the sustainability of having undergraduate healthcare students as health insurance advocates.

The main limitation of this study is that the survey was self-reported; hence, the ratings of eHealth literacy and health insurance knowledge may have been influenced by social desirability bias or self-enhancement bias. Fully anonymizing the survey and using widely validated tools reduced the risk of biases, although partially because this study leveraged previous validations instead of validating the tools for use among undergraduate healthcare students in Kenya. Another limitation is that the sample was not proportionately distributed across all the institutions of higher learning in Kenya due to the self-selection sampling, whereby individuals chose to participate instead of being selected into the study. Hence, the observed trends apply among Moi University, Meru University of Science and Technology, Kenya Medical Training College (KMTC), and Maseno University students in healthcare academic programs, which individually had sizeable representations in the sample compared to other universities. Nevertheless, the results for other institutions were similar to the overall findings, indicating that the study points to a countrywide trend of eHealth literacy and health insurance knowledge among undergraduate healthcare students.

## 5. Conclusions

Despite the high penetration rate of smartphones and the internet in Kenya, only about half of students in healthcare academic programs have high eHealth literacy levels. Also, only about a fifth of them have high knowledge levels of SHA despite the intensive and extensive mainstream and digital media campaigns to promote SHA. Since eHealth literacy emerged as the main predictor of SHA knowledge, it can be improved among undergraduate healthcare students to enhance their capacity as health insurance ambassadors. Qualitative research is required to determine why pursuing academic programs like nursing and public health predicts SHA/SHIF knowledge better than pursuing MBChB.

## Figures and Tables

**Figure 1 ijerph-22-01560-f001:**
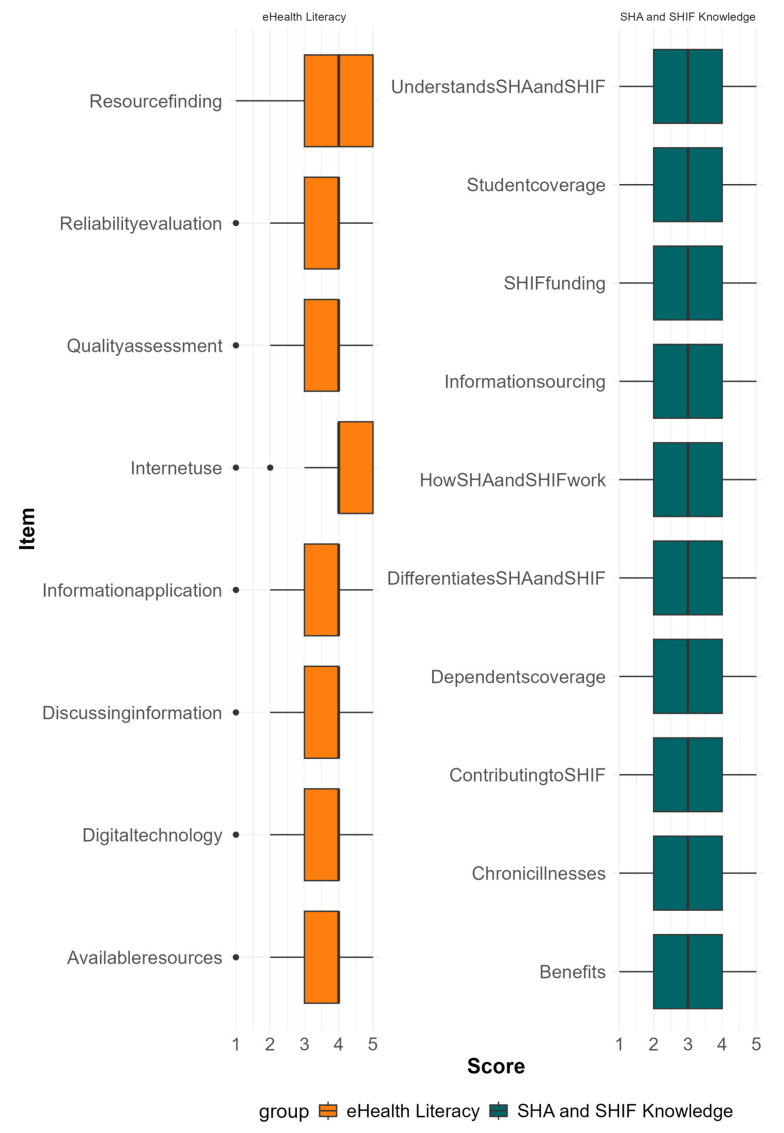
Box plots of the median ratings and quartiles of the distinct items of eHealth literacy and SHA and SHIF knowledge scales based on ratings by undergraduate healthcare students in Kenya.

**Figure 2 ijerph-22-01560-f002:**
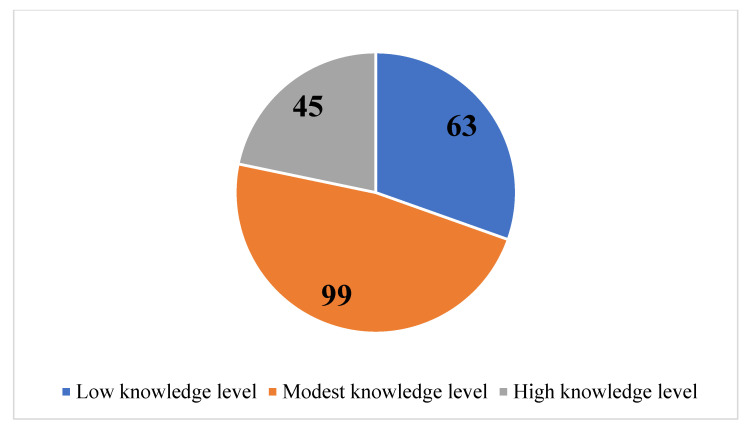
Number of undergraduate healthcare students in Kenya who self-reported their knowledge of SHA as low, modest, or high.

**Figure 3 ijerph-22-01560-f003:**
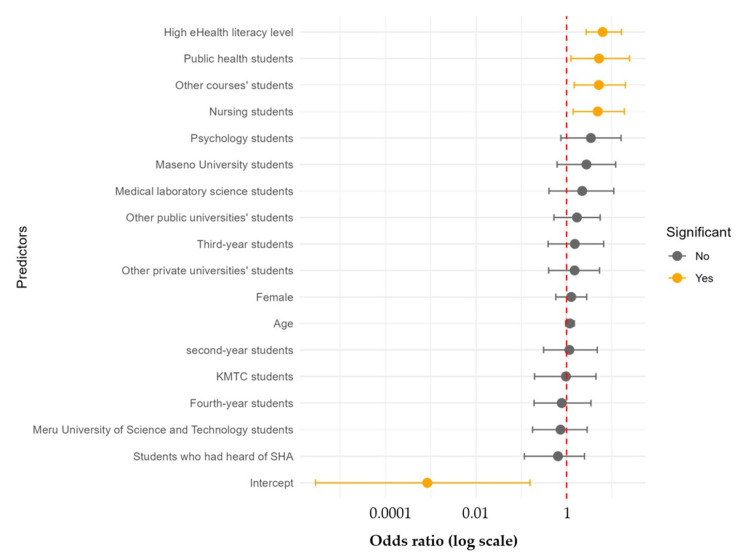
A forest plot of the odds ratios of predictors of SHA knowledge levels among undergraduate healthcare students in Kenya estimated using penalized logistic regression (the red dashed line is the odds ratio where the predictor is not statistically significantly associatiated with the outcome).

**Table 1 ijerph-22-01560-t001:** Demographic characteristics of undergraduate healthcare students in Kenya who participated in the study on eHealth literacy and knowledge of SHA/SHIF.

Characteristic	Description	Count (Percentage)
Gender	Male	112 (54.1)
	Female	95 (45.9)
Institution	Moi University	79 (38.2)
	Meru University of Science and Technology	38 (18.4)
	Kenya Medical Training College (KMTC)	21 (10.1)
	Maseno University	15 (7.2)
	Other public universities (*n* = 9)	29 (14.0)
	Other private universities (*n* = 9)	25 (12.1)
Program of study	Bachelor of Medicine and Bachelor of Surgery (MBChB)	49 (23.7)
	Nursing	31 (15.0)
	Medical Laboratory Science (MLS)	25 (12.1)
	Psychology	23 (11.1)
	Public health	18 (8.7)
	Community health	16 (7.7)
	Other	45 (21.7)
Year of study	Fourth, fifth, and sixth (senior students)	34 (16.4)
	Third	55 (26.6)
	Second	50 (24.2)
	First	34 (16.4)

**Table 2 ijerph-22-01560-t002:** Results of penalized logistic regression model to identify determinants of SHA knowledge level.

Predictor	Odds Ratio	*p*-Value
eHealth literacy	6.2	<0.0001
Nursing	4.9	0.013
Public health	5.0	0.024
Other academic programs	5.1	0.010

## Data Availability

The original data presented in the study are openly available in Mendeley Data at https:/doi.org/10.17632/t86f562rkd.1.

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
