# Peer review of "eHealth Literacy and Knowledge of Social Health Financing Among Undergraduate Healthcare Students in Kenya"

_ijerph, 2025, doi:10.3390/ijerph22101560_

Round 1

Reviewer 1 Report (Previous Reviewer 1)

Comments and Suggestions for Authors

Thank you for your revisions and comments. I see an improvement in your article after the revisions.

About the OR that is used in the Cross-sectional Study, please see the use of the OR  in the link https://pmc.ncbi.nlm.nih.gov/articles/PMC7850067/

Thank you

Author Response

Thank you for the review. We have addressed your comments in the response matrix and the manuscript. 

Reviewer 2 Report (Previous Reviewer 3)

Comments and Suggestions for Authors

The revised version incorporated practically all reviewers' objections and comments. The authors responded in detail to all the reviewers’ remarks and implemented them into the manuscript, which resulted in substantial improvement of the manuscript. The only remaining shortcoming is still in study limitations, where “the convenience sampling whereby individuals chose to participate instead of getting selected into the study” was mentioned. This was not a convenience sample but a self-selection sample, i.e. non-probability one and thus the results might not be generalized to either within the mentioned universities, whose students participated largely. Generally, the manuscript deserves to be published.

Author Response

Thank you for the review. We have addressed your comments in the response matrix and the manuscript. 

Reviewer 3 Report (New Reviewer)

Comments and Suggestions for Authors

In introduction, refer in greater detail to the concept of health literacy. This is a construct of great relevance in the study, however, there is no more in-depth approach to the different theoretical models that have tried to understand this process in a multidimensional way.

In methodology, it is suggested that a definition be made regarding the type of sampling used. As far as possible, characterize the measurement instruments used, and if possible, complement it with more psychometric information.

The results are adequately presented and allow a correct understanding of the data presented.

In relation to the discussion. The type of sampling used, and the size of the sample is limited and does not ensure the possibility of generalization in the universities to which the students belong, nor in the country in general. In addition to the above, it is considered relevant to point out the limitations of the study, but at the same time the projections that could be addressed from other studies. 

Author Response

Thank you for the review. We have addressed your comments in the response matrix and the manuscript. 

Round 2

Reviewer 1 Report (Previous Reviewer 1)

Comments and Suggestions for Authors

Thank  you for the correction.

This manuscript is a resubmission of an earlier submission. The following is a list of the peer review reports and author responses from that submission.

Round 1

Reviewer 1 Report

Comments and Suggestions for Authors

Please find my comments in the file

Author Response

Dear reviewers,

Thank you for reviewing our manuscript. Your feedback has helped us greatly strengthen the scientific communication of the findings of the study we conducted. We have edited the manuscript by addressing your comments point by point as shown in the following matrix.

We highly appreciate your contribution.  

Comment

Action

Line number

Title :

What does “predict” mean in the title?

Please make it clear in the method section

We have edited the title to reflect that the focus was on statistically testing the association between eHealth literacy and SHA/SHIF knowledge. We used ordinal logistical regression in the statistical analysis, our aim was not to create a predictive model.

2-3

1. What are the actual problems in this research? Low participation in health insurance or any other problems. I did not find a real problem in my research. Please state it first!

We have clarified the problem as low contributions into SHIF due to inactivity among Kenyans in the informal sector.

We have brought a paragraph clarifying the problem further closer.

38-39

40-51

In lines 79-82, authors wrote “Literature review did not identify studies on the eHealth literacy of health students in Kenya and how it influences their knowledge of health insurance, yet health students require the least resources to train into health insurance crusaders when their eHealth literacy is high.” What is the relevance of this research?

We have edited the statement to clarify the justification for this study by deleting the phrase “when their eHealth literacy is high” to remove the impression that the eHealth literacy is high in the background.

86-90

In lines 82-84, the authors wrote “The health students could be focal points to educate their community members about health insurance to address the characterized problem of most Kenyans having limited knowledge of the social health insurance benefits package.” What is the relevance to the research problem? Why were health students chosen as research objects?

We have edited the entire paragraph to clarify that health students were chosen as research participants because they have background health insurance knowledge and higher eHealth literacy levels compared to the general population in Kenya due to their training and access to technologies. Hence, they can be low-cost focal points for improving their communities’ knowledge of health insurance if contextually prepared.

80-90

The aim of this research was to determine the relationships or to predict?

We have edited the title and the aim to clarify that this study determined relationships; it did not develop a prediction model. 

2-3

90-94

I did not find the method to build a prediction model in the Method section

We did not build a prediction model. The logistic regression in the statistical analysis was meant to identify predictors of SHA/SHIF knowledge status from a set of independent variables, that is why the term “predictors” appeared in the manuscript. We have edited the sentence to remove the confusion.

148-149

Please add the formula that was used in the sampling method and complete it with the reference

We used G*Power 3.1.9.7 software to estimate sample size. We have added the previously omitted inputs for conducting the test. As explained in the article in this link, G*Power is a reliable tool for sample size estimation in research: https://doi.org/10.3352/jeehp.2021.18.17. “G*Power is recommended for sample size and power calculations for various statistical methods (F, t, χ2, Z, and exact tests), because it is easy to use and free.”

101-106

In lines 82-84, the authors provided information about the internal consistency test, but there was no information about the results of the validation test. Is this a limitation of this study?

That is a limitation of this study, we have acknowledged it.

We depended on research reports that eHealth Literacy Scale and the Health Insurance Knowledge Quiz have been severally validated in multiple contexts to use the tools. We have added some information about the validations of the two tools.

335-337

127, 133

There is no information about the research variables as reported in the results

We have added a paragraph on the dependent and independent variables.

122-124

What is the urgency of Figure 1? Is it the included research variable in this study?

We have deleted figure 1. Indeed no information is lost upon its deletion

In lines 189-190, the authors wrote “The model was sta s cally significant for the course (χ² (1) = 9.9, p = 0.002) and age (χ² (1) = 5.0, p = 0.03).” What is the model? Is this a sta s cal model or any other model?

We have specified that it is a logistic regression model, which is a statistical analysis approach.

203

The study design was cross-sec onal, but the author used “OR” in the statistical analysis. Is this right? The “OR” is suitable for a case-control study design.

Odds ratio is used in case control studies and also in analytical cross-sectional studies to examine the association between a factor and outcome, which is what we did in this study. DOI: 10.1097/01.NAJ.0000794280.73744.fe

The results show that there are many statistical methods for their indicators. Please explain the steps of the statistical method in the methods section.

We have enriched the data analysis subsection to describe all the statistical tests conducted and reported in the results.

138-149

Are there any discussions about the implementation of the prediction model that was found in this study

We did not find a prediction model, we only used logistical regression for the data analysis. We have edited the phrase that was confusing in the title. Sorry for the confusion.

2-3

Please provide a concise conclusion regarding the aim of this study.

We have rewritten the conclusion to precisely align with the aim of the study.

348-355

What are the recommendations based on the prediction model found in this study?

There was no prediction model in this study; the logistic regression model was misunderstood as a prediction model but we have revised the manuscript to avoid such confusion.

Reviewer 2 Report

Comments and Suggestions for Authors
  • Title is too long. It should maximum 10 - 15 words and comprehensive.
  • "The median eHealth literacy score was 3.9/5, with about 54% 22
    having high eHealth literacy levels (median ≥ 4). About 9.2% of the students had never 23
    heard of SHA/SHIF while only 21.7% had high knowledge of SHA/SHIF (median ≥ 4), 24
    with the median knowledge score being 3/5." What about remaining 15.1%?
  • Using abbreviations after defining them is a good practice. e.g. MBChB
  • Formatting should be according to the criteria given on website
  • Materials and Methods section is too comprehensive and needs more explanation.
  • Figure 1 is irrelevant
  • Figure 2 is not readable
  • Sections do not present clear study targets.

Author Response

Dear reviewers,

Thank you for reviewing our manuscript. Your feedback has helped us greatly strengthen the scientific communication of the findings of the study we conducted. We have edited the manuscript by addressing your comments point by point as shown in the following matrix.

We highly appreciate your contribution.  

Comment

Action

Line number

Title is too long. It should maximum 10 - 15 words and comprehensive.

We have shorted the title, it is now 14 words

2-3

"The median eHealth literacy score was 3.9/5, with about 54% having high eHealth literacy levels (median ≥ 4). About 9.2% of the students had never

heard of SHA/SHIF while only 21.7% had high knowledge of SHA/SHIF (median ≥ 4),

with the median knowledge score being 3/5." What about remaining 15.1%?

These statements are highlights of the major findings (high and low ratings).

For eHealth literacy, 53.5% had high eHealth literacy and 5.3% had low eHealth literacy, hence the remaining 41.1% had modest eHealth literacy.

For SHA/SHIF, 21.7% had high knowledge level of SHA/SHIF, 30.4% had low knowledge of SHA, hence the remaining 47.9% had modest knowledge of SHA.

173

196

Using abbreviations after defining them is a good practice. e.g. MBChB

We have defined all abbreviations at the first instance of mentioning them.

25, 161

Formatting should be according to the criteria given on website

We have formatted the manuscript as per the journal’s criteria.

Materials and Methods section is too comprehensive and needs more explanation

We have added explanations in the materials and methods section to enhance comprehensiveness

138-143

Figure 1 is irrelevant

We have deleted figure 1

Figure 2 is not readable

We have increased the font size of the text in figure 2 to improve readability.

Figure 2

Sections do not present clear study targets.

We have clarified the study aim, which we have demonstrated how it was addressed in the research in the conclusion.

90-94

332-339

Reviewer 3 Report

Comments and Suggestions for Authors

An interesting analytical cross-sectional study evaluating eHealth literacy and health insurance knowledge among undergraduate healthcare students in Kenya. The authors employed sophisticated methods for analyzing self-reported data obtained from an online survey. They prepared a well-structured article, which, unfortunately, still has several drawbacks needing improvement. Apart from the major corrections needed in the summary, conclusions, and keywords, the manuscript is in places sloppily written, particularly when it comes to the introduction of abbreviations, description of outcomes, and the figures’ captions/legends.

Major drawbacks

The Abstract and the Conclusions need to be rewritten. Particularly, the sentence “The median eHealth literacy score was 3.9/5, with about 54% having high eHealth literacy levels (median ≥ 4).” seems senseless (?), and even worse is the continuation (lines 23-28).

The Conclusions must summarize the study’s results; the present ones failed to do this.

The study limitations must be expanded/enhanced (lines 310-319): an important study limitation is non-probability sampling due to self-selection, as individuals choose to participate in the online study by themselves.

Several keywords (lines 31-32) are not adequate: internet and knowledge are too general, while NHIF, Social Health Authority, and Social Health Insurance Fund are too specific. 
Suggestion: Please consult MeSH (Medical Subject Headings) for appropriate keywords. 

Minor drawbacks

The terms “students in health-related courses” (line 18, 52-53, 89 and elsewhere), health courses (line 21), and healthcare courses (line 146) are not appropriate.

Suggestion: Please use “students enrolled in healthcare programs” or “undergraduate healthcare students” as replacements, since this refers to healthcare study programs rather than specific (separate) courses.

In many places, the abbreviations are used without introducing them (e.g. MBChB in line 27). Moreover, most abbreviations are listed in the Abbreviations list (355 onwards), while some are omitted from that list (e.g.  MLS used in line 193).

Suggestion: Please introduce the used abbreviations (KMTC, MBChB, and MLS) into Table 1 accordingly.

Figure 1 (word cloud for other healthcare programs) is not functional; please omit it completely, it can be replaced with a single legend below Table 1 listing the exact numbers of students enrolled in the remaining 12 courses.

The authors’ affiliations list includes three institutions (lines 7-11), but only two of them are linked to the authors’ names (line 6).

It seems that eHealth literacy levels (high, modest, and low) are not defined anywhere (?).

There are four categories of “Year of study” according to Table 2 (first, second, third, and Fourth, fifth, and sixth combined). Why then year 4’s and 3’s are mentioned (see lines 169-170). Please explain.

“Interquartile range = 5” (line 161) is imprecise. Specifically, in this case, when median=31, this could mean interquartile range = 26-31, or 27-32, …, or 31-36. Similarly, IQR=14 (line 181) could vary from 26-30 to 30-44. Moreover, the caption of Figure 2 needs to be improved (“Box plots of the median ratings of the various aspects”, it is going about box plots for distinct items of eHealth literacy and SHA/SHIF knowledge scales). Moreover, the legend of Figure 2 is incomplete (it should mention that the medians accompanied by the quartiles are shown on the graph).

Reporting of the statistical tests’ results has to be improved, too. There are references to p-values <0.0001 (Table 2, line 224-225), and p=0.0001 (line 201) instead of p<0.001, as is common. Despite very large contingency tables were analyzed with a chi-square test (3*6 table in case of crosstabulation of eHealth literacy level and healthcare study program and 3*4 table crosstabulating eHealth literacy and year of study, see lines 172-173), statistically significant values are reported (p=0.01 and p=0.02, see lines 172-173). Please correct either the chi-square value or the significance level for the first one because the significance level for df=10 and alpha=0.01 is 23.209, while the reported chi-square here is 22.1<23.209.

Comments on the Quality of English Language

The main drawback of the quality of English has already been mentioned in Comments and Suggestions for Authors above: it is the use of the term health course (courses) for healthcare study programs. Otherwise, the Quality of the English language is more or less good.

Author Response

Dear reviewers,

Thank you for reviewing our manuscript. Your feedback has helped us greatly strengthen the scientific communication of the findings of the study we conducted. We have edited the manuscript by addressing your comments point by point as shown in the following matrix.

We highly appreciate your contribution.  

Comment

Action

Line number

The Abstract and the Conclusions need to be rewritten. Particularly, the sentence “The median eHealth literacy score was 3.9/5, with about 54% having high eHealth literacy levels (median ≥ 4).” seems senseless (?), and even worse is the continuation (lines 23-28).

We have rewritten the abstract and the conclusion to make more sense.

13-28

348-355

The Conclusions must summarize the study’s results; the present ones failed to do this.

We have rewritten the conclusion to summarize the study results

348-355

The study limitations must be expanded/enhanced (lines 310-319): an important study limitation is non-probability sampling due to self-selection, as individuals choose to participate in the online study by themselves.

We have expanded the limitations section to include convenience sampling.

332-346

Several keywords (lines 31-32) are not adequate: internet and knowledge are too general, while NHIF, Social Health Authority, and Social Health Insurance Fund are too specific.

Suggestion: Please consult MeSH (Medical Subject Headings) for appropriate keywords.

We have consulted MeSH to replace the keywords with the most appropriate ones.

29-30

The terms “students in health-related courses” (line 18, 52-53, 89 and elsewhere), health courses (line 21), and healthcare courses (line 146) are not appropriate.

Suggestion: Please use “students enrolled in healthcare programs” or “undergraduate healthcare students” as replacements, since this refers to healthcare study programs rather than specific (separate) courses.

We have replaced all “health students” with “undergraduate healthcare students” or “students enrolled in healthcare academic programs” and “courses” with “academic programs.”

Throughout

In many places, the abbreviations are used without introducing them (e.g. MBChB in line 27). Moreover, most abbreviations are listed in the Abbreviations list (355 onwards), while some are omitted from that list (e.g.  MLS used in line 193).

Suggestion: Please introduce the used abbreviations (KMTC, MBChB, and MLS) into Table 1 accordingly.

We have introduced descriptions of all the abbreviations at the first mention and included all of them in the table of abbreviations

25, 161, 374

Figure 1 (word cloud for other healthcare programs) is not functional; please omit it completely, it can be replaced with a single legend below Table 1 listing the exact numbers of students enrolled in the remaining 12 courses.

We have deleted the word cloud and the 12 other academic programs since the information does not add value.

The authors’ affiliations list includes three institutions (lines 7-11), but only two of them are linked to the authors’ names (line 6).

We have linked all the three institutions to the respective authors.

4

It seems that eHealth literacy levels (high, modest, and low) are not defined anywhere (?).

We have defined them

173-177

There are four categories of “Year of study” according to Table 2 (first, second, third, and Fourth, fifth, and sixth combined). Why then year 4’s and 3’s are mentioned (see lines 169-170). Please explain.

We have corrected the mistake by renaming the students in different academic years as in table 2: first, second, third, and Fourth, fifth, and sixth combined

179-186

“Interquartile range = 5” (line 161) is imprecise. Specifically, in this case, when median=31, this could mean interquartile range = 26-31, or 27-32, …, or 31-36. Similarly, IQR=14 (line 181) could vary from 26-30 to 30-44.

Moreover, the caption of Figure 2 needs to be improved (“Box plots of the median ratings of the various aspects”, it is going about box plots for distinct items of eHealth literacy and SHA/SHIF knowledge scales). Moreover, the legend of Figure 2 is incomplete (it should mention that the medians accompanied by the quartiles are shown on the graph).

We have substituted the IQR with lower and upper quartiles for specificity.

We have edited the caption for improvement and completeness

172-194

190-192

Reporting of the statistical tests’ results has to be improved, too. There are references to p-values <0.0001 (Table 2, line 224-225), and p=0.0001 (line 201) instead of p<0.001, as is common.

Despite very large contingency tables were analyzed with a chi-square test (3*6 table in case of crosstabulation of eHealth literacy level and healthcare study program and 3*4 table crosstabulating eHealth literacy and year of study, see lines 172-173), statistically significant values are reported (p=0.01 and p=0.02, see lines 172-173). Please correct either the chi-square value or the significance level for the first one because the significance level for df=10 and alpha=0.01 is 23.209, while the reported chi-square here is 22.1<23.209.

The p-value we reported as <0.0001 was very small, way less than 0.0001, that is why we did not report the exact figure. The p-value we reported as p=0.0001 was exactly that (not less than 0.0001), that is why we reported it.

The alpha used is 0.05, not 0.01.